# Value-Based Health Care for Prostate Cancer Centers by Implementing Specific Key Performance Indicators Using a Balanced Score Card

**DOI:** 10.3390/healthcare12100991

**Published:** 2024-05-11

**Authors:** Jan Philipp Radtke, Peter Albers, Boris A. Hadaschik, Markus Graefen, Christian P. Meyer, Björn Behr, Stephan Nüesch

**Affiliations:** 1Department of Urology, Medical Faculty, Heinrich-Heine-University Düsseldorf, 40225 Düsseldorf, Germany; peter.albers@med.uni-duesseldorf.de; 2Department of Radiology, German Cancer Research Center (dkfz), 69120 Heidelberg, Germany; 3Division of Personalized Early Detection of Prostate Cancer, German Cancer Research Center (dkfz), 69120 Heidelberg, Germany; 4Department of Urology, University of Duisburg-Essen, 45147 Essen, Germany; boris.hadaschik@uk-essen.de; 5German Cancer Consortium (dktk), University Hospital Essen, 45147 Essen, Germany; 6Martini Clinic, University Hospital Hamburg-Eppendorf, 20251 Hamburg, Germany; graefen@uke.de; 7Department of Urology, Klinikum Herford, University Hospital of Ruhr-University, 44789 Herford, Germany; christian.meyer@klinikum-herford.de; 8Department of Plastic and Hand Surgery, BG-University Hospital Bergmannsheil Ruhr University Bochum, 44789 Bochum, Germany; bjorn.behr@rub.de; 9Center for Management, School of Business and Economics, University of Muenster, 48149 Muenster, Germany; stephan.nueesch@unifr.ch; 10International Institute of Management in Technology, University of Fribourg, 1700 Fribourg, Switzerland

**Keywords:** prostate cancer, Balanced Scorecard, standardized quality measurements, ICHOM, metrics, value-based healthcare

## Abstract

Background: Prostate cancer (PC) is the most common cancer in men in 112 countries, and accounts for 15% of cancers. Because it cannot be prevented, the rise in cases is inevitable, and improvements in diagnostic pathways and treatments are needed, as there is still a shortage of cost-effective diagnostics and widespread oncologically safe treatment options with measurable quality. As part of the implementation of a Full Cycle of Care, instruments have been developed to achieve value-based medicine, such as consistent commitment to measurability. One of these instruments is the Balanced Scorecard (BSC). Here, we propose the first BSC for prostate cancer (PC) treatment. Methods: BSCs are used to assess performance in healthcare organizations across four dimensions: financial, patient and referrer, process, and learning and development. This study aimed to identify Key Performance Indicators (KPIs) for each perspective. A systematic literature search was conducted according to PRISMA guidelines using multiple databases and specific search terms to identify KPIs for PC care, excluding case reports and conference abstracts. In total, 44 reports were included in analyses and development of the PC-specific BSC. Results: In the present study, a PC-specific BSC and KPIs were defined for the four classic perspectives, as well as for a newly developed PC-Specific Disease and Outcome perspective, including patient-related parameters from the German Cancer Society and the International Consortium for Health Outcomes Measurement. In addition, the Process perspective includes KPIs of fulfillment of continuing education of residents and the metrics of structured training of the radical prostatectomy procedure in the Learning and Development perspective. Conclusions: The developed BSC provides a comprehensive set of perspectives for an Integrated Practice Unit or center in PC care, ensuring that the indicators remain manageable and applicable. The BSC facilitates value creation in line with Porter’s Full Cycle of Care by systematically collecting and providing economic, personnel, and medical results, actions, and indicators. In particular, this BSC includes KPIs of structured training of practitioners and metrics of the German Cancer Society, that recently proved to improve PC patients outcomes.

## 1. Introduction

In an ideal scenario, hospitals and medical facilities should optimize their medical and non-medical structures to break even while ensuring high-quality medical care [1]. Additionally, they should implement patient-oriented services and processes, all the while maintaining satisfied and motivated employees [1,2]. Medical institutions that adhere to these principles would embody an ideal institution within an ethical, patient-centered, and economical healthcare system [1,3,4].

Therefore, it is the goal of medical institutions to focus on the various stakeholders of public facilities while continuing to fulfill their legal obligation of providing healthcare. Moreover, they must contend with internal factors such as dwindling financial resources and new employee demands. Porter presents the Full Cycle of Care as a solution to this predicament [5,6], reducing costs by improving quality. Porter argues that competition should be (i) unrestricted, (ii) based on outcomes, and (iii) centered around diseases [5,6,7,8]. The value created in this approach is characterized by expertise, experience of healthcare providers, and the specific quality related to the disease [5,6,7,8]. The results and associated costs of this value-based medicine must be systematically collected and made openly available [5,6,7,8].

Several instruments have been proposed to achieve the Full Cycle of Care. These can be divided into structural components such as Integrated Practice Units (IPUs), which are specialized departments focused on diseases and patients, spanning across subunits of the organization, and a consistent commitment to measuring outcomes through benchmarking [9,10,11]. This approach is particularly useful in the context of oncological diseases, which are often very expensive, such as novel immunotherapies, and require the highest possible quality in terms of the most significant endpoint, overall survival [10].

One of these instruments is the Balanced Scorecard (BSC), an organizational tool that is constantly evolving and acquiring new applications for implementing strategies in institutions [4,12]. The strength of BSCs lies in the integration of multiple perspectives while considering both monetary and non-monetary elements, which especially reflects the circumstances of public institutions such as university hospitals [4].

In the early 2000s, analysis of international data suggested that cancer survival rates in Germany lagged behind survival rates in other European countries [13]. These findings motivated the German Cancer Society (DKG) to establish criteria for certified cancer centers based on national guidelines. Certification started with breast cancer centers, and today, DKG certification programs are in effect for the most prevalent cancers, including prostate cancer (PC) [14,15]. Criteria for certification include, but are not limited to, surgical and radiotherapeutic expertise, staffing, psychosocial care, and minimum case numbers [16].

On the other hand, centers without certification can also diagnose and treat patients with PC. However, tumor-specific peri- and short-term postoperative patient characteristics tend to be improved in certified PC centers, whereas there is no evidence of an improvement in long-term oncological and functional outcomes [17].

Prostate cancer is not preventable [18]. The only effective way to mitigate harm is to implement strategies for early diagnosis and effective treatment. On the other hand, most men treated for non-metastatic PC do not die from PC. For example, in the UK, around 50,000 men are diagnosed with PC annually and around 12,000 men die from the disease [18]. Overall, around 80% of patients survive 10 years or more. Therefore, men often live for many years—even decades—from diagnosis, with the consequences of treatments such as surgery or radiotherapy. Due to the relatively low mortality rates, patients are at a higher risk of experiencing significant functional and psychological impairments resulting from treatment side effects. Consequently, the importance of functional outcomes, such as urinary continence and erectile function, has increased in patients’ perception of treatment quality. Interestingly, these functional outcomes are not mandatory criteria for PC center certification [16]. Thus, implementation and standardization of metrics are urgently needed and will substantially counteract the coming increases in PC and reduction in side effects worldwide [18]. Some efforts have been made by the International Consortium for Health Outcome Measurement (ICHOM), as well as defining variables of the certification process of the DKG [16,19,20]. However, these efforts are mostly patient- and disease-centered, but do not include process perspectives and learning and development of staff, physicians and researchers. In addition, certification processes lack financial and economic metrics.

Therefore, the aim of this work is to develop a customized and mostly comprehensive BSC, including multiple perspectives, for an academic oncological IPU specializing in the treatment of patients with localized and locally advanced PC using a systematic literature search.

## 2. Material and Methods

The BSC considers four dimensions or perspectives: financial perspective, referrer and patient perspective, process perspective, and employee perspective [2,3,12].

In recent years, quality metrics have been published primarily for oncological conditions, and there is a need for the implementation of a metric catalog that covers all perspectives and ensures reproducibility [19,21,22,23].

Two types of metrics are distinguished in BSCs: leading indicators and lagging indicators [2,3,12]. Leading indicators reflect the current development and can influence the short-term results, while lagging indicators indicate whether the goals have been achieved [2,3,12].

The combination of leading and lagging indicators offers the greatest benefit in the BSC.

The financial perspective provides information on the financial and earnings situation, the patient and referrer perspective focuses on retaining and attracting patients and referrers, the process perspective assesses key characteristics of business processes, and the learning and development perspective focuses on employee training and productivity [1].

Due to the relatively low mortality rate of PC, patients are at a higher risk of experiencing significant functional and psychological impairments resulting from treatment side effects. Consequently, functional outcomes, such as urinary continence and erectile function, have become more important to patients and measurements of those metrics have become a surrogate of treatment quality. To address these specific needs of PC care, an additional perspective called the ‘PC-Specific Disease and Outcome Perspective’ was implemented. This perspective includes case mix variables defined by the ICHOM, as well as patient-related outcomes (PROMs) and postoperative complications derived from the variables of the certification process of the DKG [19,20,24,25]. The KPIs were identified through a systematic literature search following the guidelines of the Cochrane Collaboration and Preferred Reporting Items for Systematic Reviews and Meta-Analyses (PRISMA) [26]. A systematic literature search was conducted using the databases PubMed, EMBASE, Cochrane Library CENTRAL, Google Scholar and Scopus for articles published prior to 21 December 2021. This search was not registered according to the PRISMA statement, nor was a protocol prepared. The bibliographies of the retrieved articles were also searched for relevant studies that were not included in the database search. Case reports and conference abstracts were excluded. In addition, we performed a literature search using Google including non-indexed reports. The search query included the items ‘Balanced Scorecard’, ‘BSC’, ‘health care’, ‘Key Performance Indicators’, ‘KPI’, ‘Prostate Cancer’, ‘Prostate Cancer Assessment’, and the German item ‘Prostatakarzinom’. Initial screening by article title and abstract was performed by one investigator to identify eligible studies. Potentially relevant studies were subjected to a full-text review. Manual searches of reference lists of relevant articles were also carried out to find additional studies of interest. The process is given in Figure 1 and the PRISMA checklists (Appendix A).

Study quality and the risk of bias were evaluated using the ROBINS-I tool for nonrandomized studies and RoB 2 for randomized controlled trials (RCTs) as outlined in the Cochrane handbook for systematic reviews of interventions (Appendix A) [27,28]. In the case of ROBINS-I, bias for every domain and overall was classified as low, moderate, serious, or critical (Appendix A). The presence of confounders was determined by one investigator. The ROBINS-I and RoB 2 assessments for individual studies were conducted by one investigator. A critical appraisal of the literature search was performed using the AMSTAR 2 checklist (Appendix A) [29].

The aims and metrics for the different KPIs were determined based on the references from which they were derived. Whenever needed and available, current benchmark metrics were derived from the Department of Urology at University Hospital Essen or Martini Clinic Hamburg.

## 3. Results

The KPIs for the PC BSC are stratified into five different perspectives as follows:

### 3.1. Financial Perspective

These KPIs are presented in Table 1 and Figure 2. Three indicator groups are distinguished [1,12]:Revenue growth and revenue mix;Cost reduction or productivity improvement;Utilization of assets or investment strategies.

The key metrics for revenue growth and revenue mix include the increase in target market share, effective case mix, and Case-Mix Index (CMI). The CMI is a severity index in the German disease-related group (DRG) system, describing the average severity of treated hospital cases and the associated relative economic resource expenditure, for example, by department, hospital, or region. The increase in CMI is the target reference, as it is an indicator of the severity of cases that are treated. The case mix is a surrogate for the number of cases and their severity. These KPIs are therefore helpful to continuously monitor the productivity of a PC center. Additional indicators like revenue, primary revenue and revenue growth rate are displayed in Table 1 and are surrogates for the financial performance of the center. All of these KPIs can be easily measured and are already implemented in controlling hospitals and cancer centers.

Since many parameters related to cost reduction or productivity improvement cannot be obtained from reference clinics, as they are not available, revenue ratios and provisions per case from the Medical Service of Statutory Health Insurance Funds (MDK) were used, which are available in reference hospitals [30]. The MDK revenue ratio shows the portion of revenue used for MDK provisions and is a suitable indicator for revenue and the quality of billing for inpatient cases [30]. In a nationwide comparison by PWC, the MDK revenue ratio is 2.1%. The MDK provision per case (EUR 94 in 2018) is also integrated into the BSC [30]. DRG revenues per case and their percentage increase are additional KPIs in the cost reduction or productivity improvement group. All these indicators are KPIs for financial productivity. Whereas this is not a specific need in PC care, it is of crucial importance for hospitals in Germany and worldwide. However, we acknowledge that even if these measurements have been analyzed by PWC, they are not common in daily practice.

Two suitable metrics for the utilization of assets or investment strategies are correlated with revenue (investment ratio, maintenance ratio). The maintenance ratio averaged 3.3% in the 100 reference hospitals analyzed by PWC in 2018. The PWC nationwide average for the investment ratio in reference hospitals is 13.2% [31]. These KPIs are important to monitor a center’s efforts to implement modern facilities and novel diagnostics and therapies, like radio-ligand therapy for patients, which is a global gap also in the Lancet Commission on PC [18]. However, besides those published in the PWC report, these KPIs have not been investigated broadly yet.

**Table 1 healthcare-12-00991-t001:** Key Performance Indicators from a financial perspective.

Key Performance Indicators of Financial Perspective	Benchmark	Aim	Reference
Revenue growth and revenue mix			
Increase in target market share	19%	25%	[31,32]
Case Mix	1440	Increase	[32]
Case Mix per full-time physician	116	142	[33]
Case Mix Index	1.72	2.27	[33]
Revenue and primary reimbursement	10,703,328.00	Increase	[32]
Revenue growth rate	13%	5% per Year	[32]
Contribution margin 1 in EUR	2,106,766.00		[32]
Contribution margin 2 in EUR	−869,441.00	0	[32]
Cost reduction or productivity improvement			
MDK revenue ratio	2.10%	Increase	[31]
MDK provision per case	94		[31]
Revenues per case in EUR	4806	Increase	[32]
Increase in revenues per case in %	7%	5% per Jahr	[32]
Assets or investment strategies			[32]
Maintenance ratio	3.30%	−0.60%	[31]
Investment ratio	13.20%	Increase	[31]

### 3.2. Patient and Referrer Perspective (Table 2 and Figure 2)

The Patient and Referrer perspective focuses on the needs and preferences of patients and referrers. KPIs in this perspective include patient and referrer satisfaction, proportion of recommendations, and positive feedback [34].

The Treatment and Service Characteristics Indicator Group includes KPIs such as overall quality of patient care as reported by referrers, frequency of problems related to admission procedures, accessibility, organization, cooperation, and information. Additional parameters include the frequency of problems with medical reports, diagnostic and therapeutic aspects, and public relations and educational offerings [34]. All KPIs of this group are related to the quality of care. This is not specifically a gap in PC care, but is important, as patients are treated over a long period of time due to favorable survival rates. These metrics are regularly queried by the PICKER survey and are therefore applicable.

The Referrer and Patient Loyalty Indicator Group classifies referrers into three categories:

Key referrers (20% of referrers) with high referral volume. An increase in the number of referrals per year is assumed as a benchmark [35,36].

Practitioners with suspected potential, who have low to moderate referral intensity but high-quality cases, are referred. Benchmarks for this segment have not been published [35].

Problem referrers burden the clinic with a negative contribution margin or show a negative referral trend. The goal for this segment is to avoid increasing the volume [30,35]. The referrer KPIs are important for PC care, as there is high competition due to the prevalence and a lot of PC centers that are available, although they are not regularly requested.

The Public Relations and Training Offerings Indicator Group includes six KPIs: sufficient information about personnel changes, the department’s range of services, training offerings, relevant training topics, schedule of training, and timely announcement of training [34].

The Patient and Referrer Relationships Indicator Group includes metrics such as staff friendliness and helpfulness, availability of appointments, and organization of the admission process [1,34]. These KPIs are not necessarily linked to PC care but to patient satisfaction. As mentioned before, a lot of PC centers exist and patients can choose the center in which they are treated, at least in Germany. Thus patient satisfaction is a potential factor for a patient’s decision for or against a center. Although regularly requested in the PICKER survey, it is not clear whether these metrics have a direct influence on the quality of PC care. Thus, they would need to be tested in a prospective trial.

Profitability is measured using revenue profitability and the Earnings Before Interests, Taxes, Depreciation and Amortisation (EBITDA) margin in %, adjusted for depreciation and neutralizing funding effects [1,2]. These KPIs are not specific for PC centers but are parameters of the profitability of a clinical unit. Whereas they are easy to request in the annual business report of a clinic, their potential benefit for improving PC care needs to be validated in further analyses.

The Market Share Indicator Group uses market share in a specific market to assess the market share of a unit, aiming for an increase [34].

**Table 2 healthcare-12-00991-t002:** Key Performance Indicators of Patient and Referrer.

Key Performance Indicators of Patient and Referrer Perspective	Benchmark	Aim	Reference
Treatment and service properties			
General frequency of problems in terms of overall quality	32%	Reduction	[34]
Problem frequency in admission procedures, accessibility and organization	26%	Reduction	[34]
Problem frequency in cooperation and information	62%	Reduction	[34]
Problem frequency in medical reports	24%	Reduction	[34]
Problem frequency in diagnostics and therapy	40%	Reduction	[34]
Problem frequency in public relations and continuing	57%	Reduction	[34]
Education offerings from the referrer perspective			
Referral and patient loyalty			
Number of referrals by key referrers	Referral of last year	Increase	[35]
Number of referrals by private practitioners with potential	Referral of last year	Increase	[35]
Number of referrals by problem referrers	Referral of last year	Reduction	[35]
Public Relations and Continuing Education Offer			
Frequency of problems regarding sufficient information about personnel changes in the department	86%	Reduction to 50%	[34]
Frequency of problems regarding the department’s continuing education offers	71%	Reduction to 25%	[34]
Frequency of problems regarding information on the department’s services	55%	Reduction to 25%	[34]
Frequency of problems regarding the relevance of continuing education topics for referrers	42%	Reduction <10%	[34]
Frequency of problems regarding timing of continuing education	58%	Reduction to 20%	[34]
Frequency of problems regarding timely announcement of continuing education	23%	Reduction <10%	[34]
Patient and Referrer Relations			
Frequency of Issues with Friendliness and Helpfulness of Personnel	26%	Reduction to 20%	[34]
Issues with Rapid Availability of Appointments	17%	Reduction	[34]
Frequency of Issues with Organization of Admissions Process	33%	Reduction	[34]
Profitability			
Average Profitability Rate	0.40%	0.40%	[31]
EBITDA-Quote	3.60%	3.60%	[31]
Market share			
Market share of treatment cases in the city of the center	19.00%	25%	http://www.lzg.nrw.de (accessed on 21 December 2021)
Market share of treatment cases in the state of the center	0.01%	Increase	http://www.lzg.nrw.de (accessed on 21 December 2021)

### 3.3. Process Perspective

The Process perspective in treating PC patients can be divided into four subcategories: innovation, treatment, service, and communication (Table 3 and Figure 2). KPIs in each subcategory are as follows:

#### 3.3.1. Innovation

Degree of implementation of identified preferences: a benchmark of 75% implementation is required [1]. Reduction in “time to market”: the benchmark is the reduction in time to implementation in ≥75% of proposed innovations [1]. Implementation of innovations is a clearly defined need in PC care within the Lancet Commission for PC and in Value-based health care. Thus, these KPIs are helpful in determining the process quality of implementations, but their application is not easy and it is unclear how and if they improve PC care and patient outcomes.

#### 3.3.2. Treatment

The time to appointment benchmark is not specified, whereas the proportion of kept appointments should be 90% and the time from the first appointment to RP should be 4–6 weeks [30,37]. These are metrics for patient service and process quality, as well as for an oncologically safe treatment [38]. In addition, this metric is easy to measure and established in practical settings, and has been applied often [38].

#### 3.3.3. Service

The benchmark for continence training to prevent one of the major side effects of PC treatment is set at 100% [39,40]. It has been proven to improve PC care [41].

#### 3.3.4. Communication

Completion of the final medical report should be within one week after discharge [30]. Postoperative questionnaire delivery to patients should have a 100% delivery and 75% response rate [30]. These KPIs are important for communication between physicians and patients but are not specific for PC care. Its direct impact on PC care improvement needs to be tested.

#### 3.3.5. Additional KPIs in the Process Perspective

The percentage of bed occupancy should target around 85–90% [30,42,43]. The duration of hospital stay after RP is approximately 5 days [30,42,43]. The immediate mortality rate after treatment (RP or radiation therapy) should be <1% [44]. The latter KPIs are directly related to quality and improvement in PC care, and percentage of bed occupancy is a KPI for optimal resource utilization, not PC-specific.

Regarding the average catheter duration postoperatively, the aimed range is 5–12 days, with a target of >95% fulfillment [45,46]. The readmission rate after RP is <8% [45]. These KPIs are directly related to PC quality and metrics of the German certification process for PC centers.

Two additional KPIs related to the learning and development perspective are also included in the Process perspective: Fulfillment of training curricula for resident physicians by the European Association of UrologFy (EAU) and the German Society of Urology (DGU): Specifically, the Weiterbildungscurriculum WeCU [47]. Both are related to the specific need for well-educated practitioners in PC care [18]. However, the Lancet Commission defined this gap in terms of worldwide education, an aim that cannot directly be targeted by European or German curricula. However, it is applicable in Germany and European countries.

**Table 3 healthcare-12-00991-t003:** Key Performance Indicators of the Process perspective.

Key Performance Indicators of the Process Perspective	Benchmark	Aim	Reference
Innovation			
Identify referrer and patient needs	75%	75%	[1]
“Time to market” of implementation of 75% of proposed innovations			[1]
Treatment			
Time to first consultation	14 days	14 days	[30]
Percentage of kept appointments	>90%	>90%	[37]
Time to radical prostatectomy	4–6 weeks	4–6 weeks	[30]
Service			
Patients with post-operative continence training	100%	100%	[30]
Time until dispatch of completed medical report to referrer after discharge	7 days	7 days	[30]
Shipping rate of postoperative surveys (Clavien-Dindo classification, EPIC-26 and EORTC-QLQ-25 6 months postoperatively to Patients)	100%	100%	[30]
Response rate of postoperative surveys (Clavien-Dindo classification, EPIC-26 and EORTC-QLQ-25 6 months postoperatively to Patients)	>75%	>75%	[30]
Bed Occupancy in percent	85–90%	85–90%	[44]
Length of Hospital Stay after Radical Prostatectomy	5 days	5 days	[30]
Mortality rate after Radical Prostatectomy	<1%	<1%	[44]
Catheter Duration after Radical Prostatectomy 5–12 days in % of patients	>95%	>95%	[46]
Readmission rate after Radical Prostatectomy	<8%	<8%	[45]
Fulfillment of the Continuing Education Curriculum of the EAU	100%	100%	[48]
Fulfillment of the Continuing Education Curriculum of the DGU (WeCu)	100%	100%	[47]
External communication			
Provisional Medical Report upon discharge, as a percentage of all patients	100%	100%	[30]
Weekly offering of patient education during hospital stay, in %	100%	100%	[30]
Monthly publication of the “Article of the Month”	100%	100%	[30]
Percentage of private patients	29%	>3.6%	[31]

### 3.4. Learning and Development Perspective

In this perspective, determining the results of the KPIs requires significant effort. The following metrics and their references are mentioned in this perspective (Table 4 and Figure 2).

Regarding surgical metrics for robotic Surgeons, the European Society of Robotic Surgery in Urology (ERUS) has a standardized training program for robotic surgeons treating PC, including 17 treatment steps. Each step is rated on a 5-point Likert scale, and a minimum rating of 3 is required to successfully complete each step [48]. All these KPIs are important to tackle the aim of well-educated surgeons as well as specific needs defined by the Lancet Commission. In addition, specific training is beneficial for oncological outcomes in robotic RP and has been proven to be broadly applicable [49].

For employee training and motivation, the number of trainings per employee per year is a KPI with the benchmark of three trainings per year [1]. The average training hours per year are another benchmark with 19.6 training hours per year [50]. Continuing Medical Education (CME) is required with a 100% achievement rate of the requirements set by the medical board for CME training [51]. For employee motivation, no benchmarks are defined for suggestions per employee and per team and for the implementation of improvements measured through incentives [1]. The KPI for the number of improvement suggestions submitted/number of employees is defined by the Hans-Böckler Foundation [52]. These KPIs are not specifically for PC centers and do not target needs in PC care directly. In addition, its applicability needs to be proven. However, they are in a broader sense helpful for staff education to deliver value-based health care to PC patients, a topic that has been identified as a gap in the Lancet Commission for PC [18].

In terms of employee satisfaction, loyalty, and productivity, voluntary turnover rate, employee turnover rate, average length of hospital tenure in years, and number of improvement suggestions submitted per employee are indirect KPIs without defined benchmarks related to employee satisfaction and center adherence but not directly related to PC care [1,52,53]. Additionally, sampling these metrics might be challenging and has not widely been performed yet.

Indicators for the topic of employee productivity are an increase in patient numbers and a decrease in complaint rates and Critical Incident Reporting System (CIRS) reports from one year to another [1,51,53]. Only the last indicator is linked to patient safety, as required for PC center certification at least in Germany [41].

Lastly, for academic institutions, the publication performance is measured using the number of publications, cumulative impact factor, average impact factor per publication (IPP), h-index (Hirsch index), and total amount of external funding acquired [51,54,55]. These indicators should at least achieve the previous year’s values. The last KPI in this perspective is impact points per employee, a metric that should be calculated for the entire department [51,54,55]. These indicators are important to monitor the productivity of an academic PC center. In most academic centers these easy-to-sample metrics are monitored regularly.

**Table 4 healthcare-12-00991-t004:** Key Performance Indicators of the Learning and Development perspective.

Key Performance Indicators of the Learning and Development Perspective	Benchmark	Aim	Reference
Radical Prostatectomy in surgically active Physicians			
Meeting the 17 metrics of standardized training in robotic radical prostatectomy according to ERUS requirements per operator	100%	100%	[48]
Group 1 of the ERUS standardized radical prostatectomy with set-up, patient positioning, establishment of pneumoperitonium, adhesion lysis and docking	100%	100%	[48]
Group 2: Opening peritoneum, endopelvine fasci preparation, dorsal vein plexus suturing, anterior and posterior prostate and prostatic pillar dissection, nerve-vascular bundle preparation and apical dissection of the prostate	100%	100%	[48]
Vesico-urethral anastomosis and anterior reconstruction	100%	100%	[48]
Bilateral pelvic lymph node dissection and finalization of the operation	100%	100%	[48]
Staff training			
Number of trainings per employee per year, including those specified by the DKG: Fire Protection, MANV, Data Protection, IT Security, Hygiene, Occupational Safety, Radiation Protection, 1 freely chosen training	8	≥8	[16,20]
Training hours per employee per year	19.6	≥19.6	[50]
ME trainings according to the Medical Chamber (200 h in 5 years for Medical Employees), compliance in % of the employees	100	100	Federal Medical Chamber
Employee Motivation			
Number of submitted improvement suggestions/number of employees per month	Comparison pre-month		[52]
Voluntary resignation rate	Comparison pre-year	Reduction	[52]
Fluctuation rate	Comparison pre-year	Reduction	[52]
Median length of employment of employees in years	Comparison pre-year		[52]
Employee Productivity			
Number of Primary Cases per Center	>100	>100	[16,20]
Number of Primary Cases > DKG Prostate Centers Median	>159	>159	[16,20]
Number of Radical Prostatectomies	>50	>50	[16,20]
Number of Radical Prostatectomies > DKG Prostate Centers	>79	>79	[16,20]
Number of Complaints/Number of Patients Summoned	<15%	<15%	[51]
Number of Complaints/Number of Planned Operations	Comparison pre-year	Reduction	[51]
Number of CIRS Reports	Comparison pre-year	Reduction	[51]
Publication performance			
Number of publications per medical staff member per year	Comparison pre-year	Increase	[51]
Cumulative Impact Factor per medical staff member per year	Comparison pre-year	Increase	[51]
Average Impact Factor per publication	Comparison pre-year	Increase	[51]
Hirsch-Factor per medical staff member	Comparison pre-year	Increase	[51]
Total amount of third-party funding per medical staff member	Comparison pre-year	Increase	[51]
Impact points per year/Number of medical staff members	Comparison pre-year	Increase	[51]

### 3.5. PC-Specific Disease and Outcome Perspective (Table 5 and Figure 2)

The ICHOM has defined case mix variables that are so crucial for PC that they should be collected outside of the four classic BSC perspectives [19]: micturition symptoms using the Expanded Prostate cancer Index Composite (EPIC)-26 questionnaire and preoperative sexual function using the European Organization for Research and Treatment of Cancer (EORTC) QLQ-25 questionnaire [19]. Furthermore, postoperative PROMS should be assessed using the questionnaires, as well as postoperative complications using the Clavien-Dindo and radiation therapy complications using the Common Terminology Criteria of Adverse Events (CTCAE) classification [19,25]. All these KPIs are directly related to the need to deliver value-based health care to patients suffering from local PC with oncological security and reduction in potential side effects [18,56]. They can be easily applied using questionnaires and have been demonstrated to be feasible in the improvement of side effects of PC treatment [57,58].

For each center, 16 parameters defined by the DKG are collected, some of which have been modified for this work (Figure 2) [20]. The German Cancer Aid provides guidance for interpreting the pre- and post-therapeutic micturition symptoms and sexual function. The reference values for each group should be met or exceeded within the BSC, especially postoperatively. This metric is also suitable for use as a reference for therapeutic improvements over time, provided that the questionnaire is administered pre-therapeutically and 6 months post-therapeutically. This group of KPIs directly aims to reduce potential side effects of the treatment of PC patients and has been validated several times and proved to improve at least short-term post-interventional outcomes [17,18,20].

Similarly, the EORTC QLQ-25 questionnaire should be analyzed. Complications after surgical therapy are standardized and evaluated using the Clavien-Dindo classification on a scale from I to IV. In particular, severe complications requiring surgical intervention (Grade III) or life-threatening complications (Grade IV) are of interest [25]. The median for such complications was 5.2% in the Onkozert survey conducted by the DKG in 2021 [20]. This value sets the benchmark that the center should not exceed per year. As with the group before, these KPIs are related to a reduction in the side effects of PC treatment [18,20]. These analyses demonstrated improvement of PC care when these metrics are applied [17,41,59,60].

Complications after radiation therapy are documented using the Common Terminology Criteria for Adverse Events (CTCAE). Here, again, Grades III (requiring interventional treatment) and IV (life-threatening) are particularly noteworthy. However, the DKG’s target requirement is stricter for non-surgical therapy and should be 0% per year [20]. This also supports the need to deliver high-quality PC care in terms of radiation therapy [18]. Within the validation of certified PC centers, these KPIs have been proven to be applicable and beneficial [17].

The KPIs for PC centers within the PC-Specific Disease and Outcome perspective are predefined by the DKG in the context of certification procedures. Some indicators have already been discussed in the context of staff productivity and complications after therapy. In total, 17 indicators are incorporated into the BSC (Table 5). All these KPIs monitor actions for sufficient high-quality oncological PC care with reduced side effects, which is important to the long follow-up period of these patients. They are easy to apply and have been proven to improve the quality of care [17,56].

**Table 5 healthcare-12-00991-t005:** Key Performance Indicators of the PC-Specific Disease and Outcome perspective.

Key Performance Indicators of the PC-Specific Disease and Outcome Perspective	Benchmark	Aim	Reference
No unwanted urine leakage pre- and post-therapy	92–94%/63–77%	>63–77%	[19]
Irritative/obstructive micturition symptoms pre- and post-therapy	80–87%/89–95%	≤89–95%	[19]
No gastrointestinal complaints pre- and post-therapy	92–95%/89–95%	≥89–95%	[19]
No issues concerning sexuality pre- and post-therapy	48–66%/14–30%	>14–30%	[19]
No issues regarding hormonal function and vitality (hot flashes, depression, fatigue) pre- and post-therapy	81–90%/72–87%	>72–87%	[19]
Issues with urinary function	15%	≤15%	[16,19,20]
Postoperative Incontinence	8%	≤8%	[16,19,20]
Irritative gastrointestinal symptoms 6 months post-operatively	5%	≤5%	[16,19,20]
Therapy-associated side effects after RPX	10–12%	≤10–12%	[16,19,20]
Normal sexual activity six months after RPX	35%	≥35%	[19]
Post-operative satisfactory sexual function	40%	≥40%	[19]
Clavien-Dindo-Classification Grade III/IV in % after RPX	6.50%	≤6.50%	[16,19,20]
CTCAE classification grade III/IV in% after radiation therapy	0%	0%	[16,19,20]
Number of primary cases for low, medium and high risk PC, median per year	29/51/42	>Median	[16,20]
Presentation in pre- and post-therapeutic tumor conferences	95%/100%	≥95%/100%	[16,20]
Number of low-risk carcinoma patients receiving active surveillance, median	21%	≥21%	[16,20]
Radiotherapy + hormone therapy for high-risk PC	≥75%	≥75%	[16,20]
Psychooncological care, median	≥20%	≥20%	[16,20]
Counseling by social service, median	≥50%	≥50%	[16,20]
Participation in studies, in % of patients	≥5%	≥5%	[16,20]
Rate of R1 Resections in organ confined PC	<10%	<10%	[16,20]
Percentage of patients with definitive radiotherapy per center	≥17%	≥17%	[16,20]
Radiation Therapy for Recurrent PC after RP with PSA less than 0.5 ng/mL	70%	≥70%	[16,20]

In the present work, a PC-specific BSC (Figure 2 and Figure 3) and KPIs were defined for the four classic BSC perspectives, and a newly developed PC-specific Disease and Outcome perspective for treating localized and locally advanced PC in an academic setting. Several aspects need to be discussed in the development of this BSC.

## 4. Discussion

The indicators of the PC-specific outcome mainly include PROMs from the DKG and the ICHOM (Table 5) [19,20]. These parameters were not integrated into the patient perspective, as they are specific for PC and incorporating personal somatic and psychological treatment outcomes of men suffering from PC that are not only process-specific, like in the Patient and Referrer perspective, covering service issues, loyalty and relations. However, these specific PROMs are essential for the goals of a PC BSC [4,12,61]. For example, the parameters of the EORTC-QLQ-25 reflect important aspects for patients such as disease-specific outcomes after RP (continence, sexual function, Clavien-Dindo classification) or radiotherapy (RT). The EPIC-26 metrics focus on patient-specific outcome parameters and compare pre- and postoperative conditions, with a particular interest in postoperative continence and sexual function.

The combination of outcome and disease perspectives incorporates established patient-specific and center-specific validated metrics and is suitable not only for internal quality control, such as for surgeons, but also provides the basis for successful certification as a PC center.

In the financial perspective, three groups can be distinguished in which the metrics can be integrated: revenue growth, productivity improvement or cost reduction, and investment strategy. The first KPI in the revenue growth group is market share, which was measured for two parameters: primary cases and RPs. Since primary cases may only have an initial contact at the center for consultation, RP was used as a KPI. The validity of the reference values is ensured by the data source from Germany’s Federal Joint Committee (G-BA). Unfortunately, effective case-mix data for individual centers were not available. However, since the effective case mix is a valid parameter, particularly due to the adjustments for transferred patients, it was integrated into the BSC.

Better data exist for case mix per full-time physician. This KPI has the advantage of including the effective performance and case severity of the department, but it correlates with the number of medical staff. Therefore, the KPI is also a suitable parameter to assess effectiveness in terms of case severity in association with each employee. PWC has published data in the literature representing a nationwide cross-section of 100 hospitals in Germany [31]. However, isolated urological data have not been published yet [31]. A case mix per full-time physician is available from the specialized Martini Clinic in Hamburg, which was 142 in 2016 [24,33]. Therefore, an increase compared to the previous year’s result is sought. However, it should be acknowledged that this KPI also depends on the facility [32]. In a center with emergency service, staff members are bound to this service and are potentially not able to cover complex patients. In addition, an emergency service implies that patients cannot be allocated at forehand prior to entering the center, therefore limiting the ability to select favorable patients. The other KPIs in the revenue growth group are individual contribution margins and revenue, which are highly individual parameters that cannot be meaningfully compared between different institutions. In the cost reduction and productivity improvement group, the MDK revenue ratio, MDK provision ratio, and DRG revenue per case were chosen as parameters. The benchmarks for the first two KPIs were again based on the references used in the PWC hospital comparison [31]. Overall, it can be noted that there are very few PC-specific reference values that can be used for comparison in this perspective. Therefore, comparisons over time had to be used in many cases.

The KPIs of the patient and referrer perspectives, regarding treatment and service characteristics, are mainly referenced by the PICKER Institute [34]. However, it should be noted that the results of the PICKER survey are not specific to urology or PC.

For the indicator group “Referring Physicians and Patient Loyalty”, reference values for some indicators had to be generated partially from published increases in case volume and adapted to the respective referrer classes. The profitability indicators used were the profit margin and the EBITDA margin (%). The advantage of using EBITDA over EBIT is seen in its adjustment for depreciation and neutralization of subsidies [1,31]. Both the EBITDA margin and the profit margin were referenced using data published by PWC [31]. However, the representativeness of these values from 100 nationwide hospitals needs to be discussed as they are not specific for urology or a PC IPU.

From the process perspective, the implementation level of identified desires of the patient, referring physician, and employee surveys was integrated as a KPI for innovation. However, there are no publications available. For the treatment subgroup, prostate-specific KPIs were derived from the Martini Clinic (time to first appointment: 14 days, time to RP: 4–6 weeks), and the percentage of adhered appointments for oncology centers was reported as 90% [30,37]. These reference indicators are also in line with an oncological perspective for PC and recent guidelines [62]. In terms of service, the reference values from the Martini Clinic regarding completion of medical reports, postoperative sending of questionnaires for the Clavien-Dindo classification, EPIC-26, and EORTC-QLQ-25 were adopted, which were reported as 100% for sending with a response rate of 75% [30].

The reference values for prostate-specific KPIs such as catheterization duration and mortality rate were adapted from multicenter studies with large patient collectives, giving them reference character [44,45,46]. Regarding bed occupancy rate, the optimal utilization was based on a study by Kuntz et al., which postulated an optimal bed occupancy rate of 85–90% [42]. The structural training regulations are adopted from the EAU and the DGU and should be fully complied with.

Lastly, the KPIs of the learning and development perspective are discussed. Some KPIs, such as the surgical steps of RP according to the ERUS, are highly specific to PC and intended for medical or surgically active staff, while others are more global in terms of employee training [48]. The fulfillment of training requirements prescribed by the German Onkozert certification per employee per year is a more global KPI [20]. The reference values for KPIs related to employee motivation, loyalty, and satisfaction were particularly challenging to collect. For the latter, the methodology of data collection posed difficulties as well [52]. While there are a number of KPIs and metrics published by Havighorst regarding indirect KPIs of employee loyalty and motivation, there are no reference values available [52]. On the other hand, reference values have been published for most key figures regarding employee productivity (primary cases according to DKG specifications) [20].

In terms of patient safety, the establishment of a CIRS system was proposed, along with a decreasing number of CIRS reports from year to year. The final issue is publication performance which is specific to academic institutions. The cumulative impact factor, which can be measured per physician and per department, as well as the Hirsch factor for medical staff, is particularly noteworthy [51,54,55].

One issue of the implementation of this novel BSC is its applicability in healthcare systems outside Germany. We acknowledge that not all perspectives of the BSC are applicable. In particular, nearly all KPIs from the financial perspective are only useful in the German healthcare system, besides assets or investment strategies. From the patient and referrer perspective, profitability and market share KPIs are useful outside Germany. However, KPIs that are associated with referrers are only applicable in countries and healthcare systems with referrers. On the other hand, treatment and service KPIs are adoptable in various countries. Service and communication KPIs of the Process perspective are not specific to one healthcare system. However, innovation and treatment indicators are not useful in centralized healthcare systems.

The indicators and benchmarks for the Learning and Development perspective regarding the education of RP surgeons, publication performance, employee motivation and staff training are not directly related to any healthcare system, but robotic surgical systems are a prerequisite. The KPIs for employee productivity are related to the benchmarks of the German Cancer Aid. The indicators of the PC-Specific Disease and Outcome perspective are mostly derived from the German Cancer Aid and the ICHOM and are therefore applicable. In general, one advantage of a BSC is that perspectives and KPIs can be weighted differently in specific scenarios. For example, in countries without referral systems, these KPIs can be adopted by users for specific needs.

Another issue might be the adoption of clinical scenarios. Whereas KPIs of the Learning and Development phase do not only measure the performance of the individual, but are also beneficial for them, continuous monitoring of metrics like in the referrer and patient, or PC-Specific Disease and Outcome perspective are time-consuming and challenging. Three strategies to overcome these challenges might be helpful. One option would be full integration into quality management processes. Thus, users are able to recognize that the integration of the BSC or KPIs is not only time-consuming but can set the path of systematic quality management that is time-sparing in the long run. Next, one responsible user can aid in implementing the BSC in the clinic or unit. Lastly, communication of potential benefits of the entire BSC or separate perspectives and KPIs for individual employees and doctors or their education, for example, in the Learning and Development or the PC-Specific Disease and Outcome perspective, might help to improve acceptance of continuous quality measurements.

In addition to the limitations of partially constructed reference values for some indicators, it should be noted that, besides the systematic literature search that was performed according to PRISMA criteria, the present BSC does not claim to be exhaustive [26,29]. Furthermore, to date, there are no existing BSCs for PC IPUs, and the weighting in favor or against individual parameters can only be objectively determined based on existing literature.

Secondly, there might be some parameters that may subjectively appear equally important and worth mentioning but were not integrated. For example, from the PC-Specific Disease and Outcome perspective, a normal sexual function after six months was integrated. However, there is evidence that sexual function might improve up to one and a half years after RP. Thus, some users of the BSC might prefer the metric of sexual function after one or two years. A second example might be the rate of successful catheter removal on days 5–12 after RP. However, some centers remove the catheter after three days. Thus, the KPI needs to be adapted. Another example might be the utilization of the EORTC QLQ-25 questionnaire. However, other caregivers might prefer the EORTC QLQ-30 questionnaire or focus more on decision regret, using the Decision Regret Score or the Brief Symptom Inventory Score. However, the EORTC QLQ-25 is reliable and common, so we decided to include this score in the BSC.

Thirdly, while the KPIs are systematically categorized into the four classical BSC perspectives and the newly established PC-Specific Disease and Outcome perspective, they are not accompanied by actionable measures to achieve these goals yet. In particular, this is of interest for the PC-Specific Disease and Outcome perspective, which has not been prospectively validated as an entire set of KPIs yet. However, all KPIs integrated into this perspective have been externally validated on their own or in combination several times and are applicable and requestable in routine. 

Fourthly, we acknowledge that the proposed BSC is a comprehensive compendium of different complex perspectives in an academic PC center. In total, it embraces 103 KPIs and is therefore somewhat in contrast to the original purpose by Kaplan and Norton to minimize information overload by limiting the number of measures used [12].

Next, we have to acknowledge that the proposed BSC and its perspectives are developed for an IPU that treats local and local advanced PC. Treatment of metastatic PC with antihormonal agents and chemotherapy has other preconditions and needs that specifically do not focus on surgical or radiotherapeutic KPIs. In addition, we acknowledge that this BSC comprises both treatment options, RP and RT, but metrics tend to focus more on RP, in particular on robotic RP, rather than RT. However, treatment of local PC using RP has more performance indicators to fulfill than RT, and specific treatment metrics of RT are also covered in the PC-Specific Disease and Outcome perspective. From a practical point of view, robotic RP is more frequent in Germany and has more reproducible KPIs, like the parameters of the ERUS training course, included in the Learning and Development perspective [48].

This BSC focuses on specialized academic centers. Academic centers differ from conventional hospitals due to an extended scope of responsibilities. They are also designed to support research and education. This BSC takes that into account, in particular from the Learning and Development perspective. However, as this BSC is a comprehensive compilation, also non-academic centers and stakeholders will find KPIs that are helpful for their purposes. Thus, it cannot be a universally applicable BSC. Instead, each individual hospital must develop its own strategy-aligned BSC.

Next, we acknowledge that we included literature in this systematic review and BSC development until December 2021. This might cause a bias since the relevant literature published in 2022 and 2023 might exist, leading to different KPIs and metrics of this BSC. Lastly, limitations of the systematic review in general might result from the selection of studies, choice of relevant outcomes, methods of analysis, interpretation of heterogeneity, and generalization and application of results. In detail, only two randomized trials and eight non-randomized trials could be included in this systematic review. However, the risk of bias was judged as low for the two randomized trials that were included in this review (Appendix A). This was also the case for most of the included non-randomized studies (Appendix A). According to the AMSTAR checklist (Appendix A), we acknowledge that only one reviewer selected the eligible studies, data extraction was not performed in duplicate and we did not provide a separate list of excluded studies. On the other hand, the review was conducted according to the PICO criteria, the protocol of the systematic review and the search strategy was sufficient and the risk of bias assessment was performed as previously mentioned.

## 5. Conclusions

In summary, the developed BSC serves as an initial compendium of the diverse perspectives for a value-creating academic IPU in the diagnosis and treatment of local and local advanced PC. The various perspectives and the systematic approach of the BSC ensure that the abundance of indicators remains manageable and applicable to users. The BSC contributes to achieving the value creation envisioned by Porter’s Full Cycle of Care in value-based medicine by systematically collecting and providing access to economic, personnel, and medical results, actions, and indicators. This allows for the assessment of the current state of a PC center and identifies areas for improvement.

## Figures and Tables

**Figure 1 healthcare-12-00991-f001:**
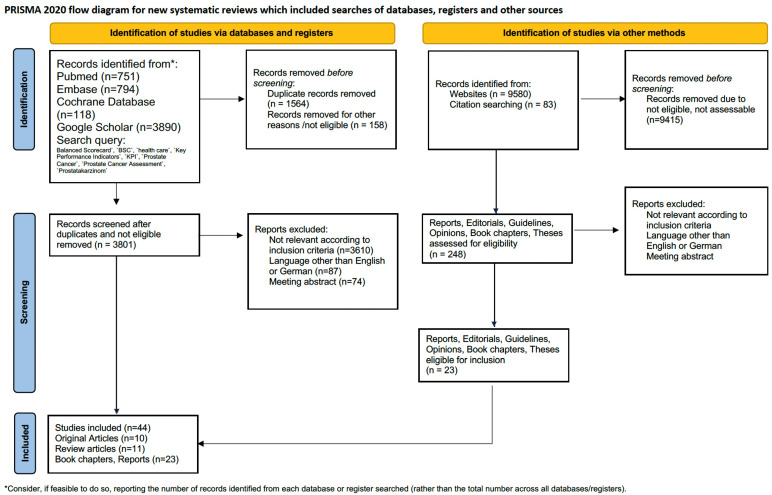
PRISMA flow diagram.

**Figure 2 healthcare-12-00991-f002:**
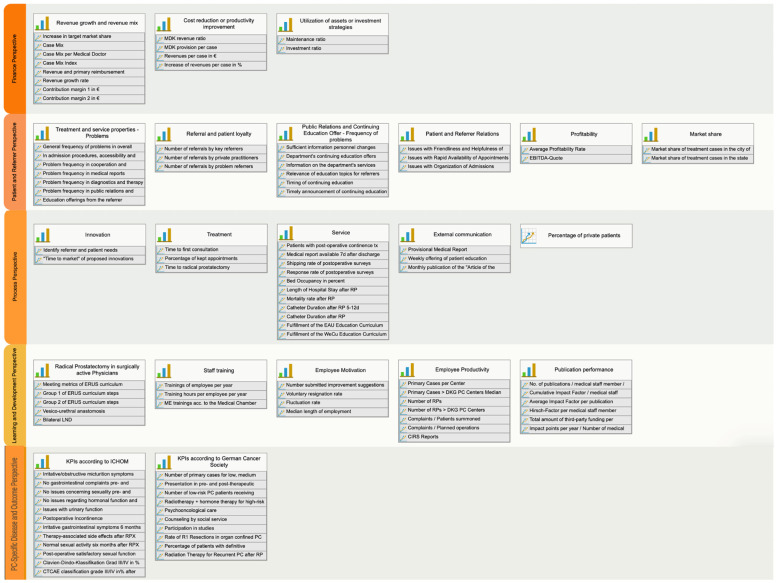
Flow chart of the developed Balanced Scorecard including KPIs in five different perspectives.

**Figure 3 healthcare-12-00991-f003:**
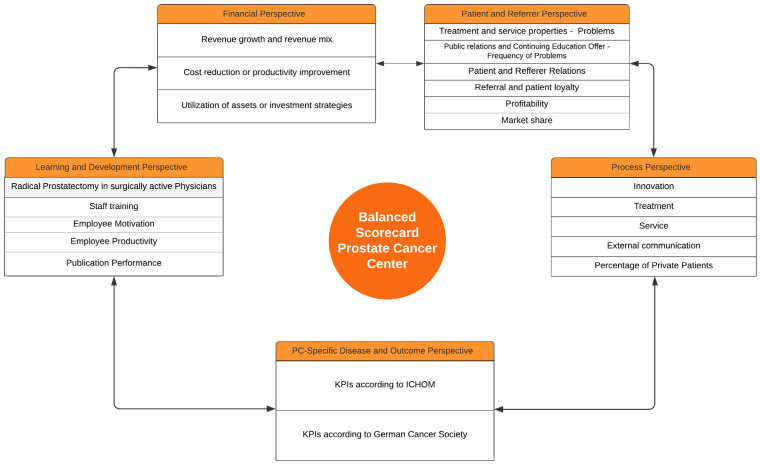
Overview of the five dimensions of the developed Balanced Scorecard.

## Data Availability

Data are contained within the article and Appendix A.

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
