# Peer review of "Value-Based Health Care for Prostate Cancer Centers by Implementing Specific Key Performance Indicators Using a Balanced Score Card"

_healthcare, 2024, doi:10.3390/healthcare12100991_

Round 1

Reviewer 1 Report

Comments and Suggestions for Authors

Overall, well written paper and important topic of improving performance and quality of care for advanced prostate cancer.

The abstract and introduction of the paper will benefit from highlighting the need to address care for advanced prostate cancer (PC). Why is it important to focus on PC—quality, costs, equity, access, etc…? The first mention of PC is the aims statement.

Page 3 states articles published from the beginning – Dec 21, 2021.  What is the beginning? Do you mean all literature prior to Dec 21, 2021?

For consistency, perhaps use “Prostate-Specific OR PC-Specific Disease and Outcome Perspective’ throughout paper and table 5, figures 2 and 3.

The authors discuss limitations of study setting, data collection, and metrics, however, potential limitations related to the systematic literature review are not discussed. 

Comments on the Quality of English Language

Minor grammatical and spelling edits.

Line 85- “The combination of leading aid lagging indicators offers the greatest benefit in the BSC.” Do you mean leading ‘aiding’ lagging?

Line 396- “Next, we have to acknowledge that the purposed BSC and its perspectives are for developed for an IPU that treats local and local advanced PC. “Do mean ‘proposed’ BSC?

Line 397- Do you mean “perspectives are developed for an IPU…”(remove ‘for’ before the word developed?)

I may have missed other instances and a thorough review will be useful.

Author Response

Responses to Reviewer #1:

Overall, well written paper and important topic of improving performance and quality of care for advanced prostate cancer.

Thank you very much for your kind and thoughtful review.

The abstract and introduction of the paper will benefit from highlighting the need to address care for advanced prostate cancer (PC). Why is it important to focus on PC—quality, costs, equity, access, etc…? The first mention of PC is the aims statement.

Thank you very much for this comment. We have included the paragraph ` Prostate cancer (PC) is the most common cancer in men in 112 countries, and accounts for 15% of cancers. Because it cannot be prevented, the rise in cases is inevitable, and improvements in diagnostic pathways and treatments are needed, as there is still a shortage in cost-effective diagnostics and widespread oncologically safe treatment options with measurable quality. As part of the implementation of a Full Cycle of Care` into the abstract (Page 4).

Page 3 states articles published from the beginning – Dec 21, 2021.  What is the beginning? Do you mean all literature prior to Dec 21, 2021?

Correct. Thank you for this clarification. We have changed this accordingly (please see Page 9).

For consistency, perhaps use “Prostate-Specific OR PC-Specific Disease and Outcome Perspective’ throughout paper and table 5, figures 2 and 3.

Thank you for this advice. We have changed this throughout the entire manuscript.

The authors discuss limitations of study setting, data collection, and metrics, however, potential limitations related to the systematic literature review are not discussed. 

Limitations of the systematic review in general might result from selection of studies, choice of relevant outcome, methods of analysis, interpretation of heterogeneity, and generalization and application of results. In detail, only two randomized trials and eight non-randomized trial could be included into this systematic review. However, the risk of bias was judged as low for the  for the two randomized trials that were included into this review, the risk of bias was judged as low (Supplementary Material 5). This was also the case for most of the included non-randomized studies (Supplementary Material 6). According to the AMSTAR checklist (Supplementary Material 7), we acknowledge that only one reviewer selected the eligible studies, data extraction was not performed in duplicate and we did not provide a separate list on excluded studies. On the other hand, the review was conducted according to the PICO criteria, the protocol of the systematic review and the search strategy were sufficient and risk of bias assessment performed as previously mentioned. We have included a paragraph on this in the limitations section of the revised manuscript (Page 30).

Comments on the Quality of English Language

Minor grammatical and spelling edits.

Line 85- “The combination of leading aid lagging indicators offers the greatest benefit in the BSC.” Do you mean leading ‘aiding’ lagging?

Thank you for reviewing this. We corrected this accordingly:` The combination of leading and lagging indicators offers the greatest benefit in the BSC.`

Line 396- “Next, we have to acknowledge that the purposed BSC and its perspectives for are developed for an IPU that treats local and local advanced PC. “Do mean ‘proposed’ BSC?

We have changed this accordingly: Next, we have to acknowledge that the proposed BSC and its perspectives are developed for an IPU that treats local and local advanced PC.` (Page 29)

Line 397- Do you mean “perspectives are developed for an IPU…”(remove ‘for’ before the word developed?)

We have changed this accordingly: Next, we have to acknowledge that the proposed BSC and its perspectives are for developed for an IPU that treats local and local advanced PC.` (Page 29)

I may have missed other instances and a thorough review will be useful.

We have reviewed the entire manuscript and attached files for quality of english language.

Reviewer 2 Report

Comments and Suggestions for Authors

Please see the attached report.

Comments on the Quality of English Language

Moderate editing of the English language is required.

Author Response

Responses to Reviewer #2:

SUMMARY This paper presents the first Balanced Scorecard (BSC) designed for prostate cancer treatment. The BSC's traditional four-dimensional assessment framework is extended with a new disease and outcome perspective. The author's innovative approach to adapting the BSC to a specific healthcare context highlights the BSC's versatility as not only an organizational tool but also its potential to improve the measurability and implementation of value-based medicine. However, I have identified a few areas that could potentially benefit from further clarification or improvement.

COMMENTS 1) Abstract: a. It is not common practice to include search keywords in the abstract. Instead, it is suggested that they be listed only under the "Method" section.

Thank you very much for this suggestion. We have changed this accordingly: ‘Methods:

BSC are used to assess performance in healthcare organizations across four dimensions: financial, patient and referrer, process, and learning and development. This study aimed to identify key performance indicators (KPIs) for each perspective.

A systematic literature search was conducted according to PRISMA guidelines using multiple databases and specific search terms to identify KPIs for PC care, excluding case reports and conference abstracts. In total, 44 reports were included into analyses and development of the PC-specific BSC.’

  1. The abstract does not require the use of abbreviations such as DKG and ICHOM, which are only used once.

We have changed this accordingly.

  1. The abstract mentions the development of a BSC specifically for PCs as the study's contribution. However, I would suggest to briefly highlight one or two KPIs that are particularly innovative or critical to the study's contributions. This would provide readers with a clearer understanding of what distinguishes this BSC from others.

Thank you very much for this suggestion. We have added information on KPIs in the process and learning and development perspectives: ‘In addition, the Process Perspective includes KPIs of fulfillment of continuing education of residents and the metrics of structured training of the radical prostatectomy procedure in the Learning and Development Perspective.’ (Page 5)

  1. The conclusion discusses how the BSC can facilitate value creation in accordance with Porter's Full Cycle of Care. However, it does not explicitly state how this implementation could enhance current practices in PC care. To improve the abstract further, I would suggest to provide a specific example of how implementing BSC has resulted in tangible improvements in PC care, such as better patient outcomes, reduced treatment costs, or increased patient satisfaction.

Thank you for this suggestion. We have added this paragraph to the conclusion of the abstract: ‘In particular this BSC includes KPIs of structured training of practitioners and metrics of the German Cancer Society, that recently proved to improve PC patients`outcomes.’

2) Introduction: The introduction should identify current challenges in implementing value-based healthcare for prostate cancer. It should also mention previous attempts to address these challenges and explain how the proposed BSC model offers a novel solution or improvement.

Thank you for this detailed review of the introduction. You are right that these specific challenges and attempts were not mentioned yet. We included a paragraph to the introduction section of the revised manuscript (Pages 8/9): ‘In the early 2000s, analysis of international data suggested that cancer survival rates in Germany lagged behind survival rates in other European countries (13). These findings motivated the German Cancer Society (DKG) to establish criteria for certified cancer centers based on national guidelines. Certification started with breast cancer centers, and today, DKG certification programs are in effect for the most prevalent cancers, including prostate cancer (PC) (14,15). Certification initially succeeded without state funding and was supported by the enthusiasm of clinics and their physicians. Criteria for certification include, but are not limited to, surgical and radiotherapeutic expertise, staffing, psychosocial care, and minimum case numbers (16).

On the other hand, also centers without certification can diagnose and treat patients with PC. However, tumour-specific peri- and short-term postoperative patient`s characteristics tend to be improved in certified PC centers, whereas there is no evidence for an improvement in long-term oncological and functional outcome (17).

Prostate cancer is not preventable (18). The only effective way to mitigate harm is to implement strategies for early diagnosis and effective treatment. On the other hand, most men treated for non­metastatic PC do not die from PC. For example, in the UK, around 50 000 men are diagnosed with PC annually and around 12 000 men die from the disease (around 7000 of whom were diagnosed with metastatic disease) (18). Overall, around 80% of patients survive 10 years or more . Therefore, men often live for many years—even decades—from diagnosis, with the consequences of treatments such as surgery or radiotherapy, however. Due to the relatively low mortality rates, patients are at a higher risk of experiencing significant functional and psychological impairments resulting from treatment side effects. Consequently, the importance of functional outcomes, such as urinary continence and erectile function, has increased in patients’ perception of treatment quality. Interestingly, these functional outcomes are not mandatory criteria for PC center certification (16). Thus, implementation and standardizations of metrics is urgently needed and will substantially counteract the coming increases in PC and reduction of side-effects worldwide (18). Some efforts have been made by the International Consortium for Health Outcome Measurement (ICHOM), as well as defining variables of the certification process of the German Cancer Society (DKG) (16,19,20). However, these efforts are mostly patient- and disease centered, but do not include process perspectives and learning and development of staff, physicians and researchers. In addition, certification processes are lacking financial and economic metrics.

Therefore, the aim of this work is to develop a customized and mostly comprehensive BSC, including multiple perspectives, for an academic oncological IPU specializing in the treatment of patients with localized and locally advanced PC using a systematic literature search.’

3) Methods In line 91, the authors state, "To address the specific needs of PC care, an additional perspective called the Prostate-Specific Disease and Outcome Perspective was implemented." However, they do not elaborate on what these specific needs are, making it unclear to the reader. A clearer identification of these needs and how they are not met by existing models would strengthen the rationale for this new perspective.

We have added a sentence to the methods section (Page 10): ‘Due to the relative low mortality rate of PC, patients are at a higher risk of experiencing significant functional and psychological impairments resulting from treatment side effects. Consequently, functional outcomes, such as urinary continence and erectile function have become more important to patients and measurements of those metrics became a surrogate of treatment quality.  To address these specific needs of PC care, an additional perspective called the `PC-specific Disease and Outcome Perspective` was implemented.’

4) Results a. I have noticed that the results section, particularly section 3.5, lacks clarity and coherence, which is the paper's main contribution. The authors should provide precise definitions for each KPI, establish relevance to the subject, and present a clear rationale for how each KPI directly addresses the identified gaps in prostate cancer care. It is also important to detail how these KPIs are integrated into the broader BSC framework for a comprehensive understanding.

Thank you for the detailed review in this section. We have revised the complete results section, provide definitions and relevance of all indicators in the context of the specific aims of a BSC for PC care. For details please see Pages 14-28.

  1. A critical aspect of this new approach is its potential to enhance prostate cancer care and outcomes. The article must demonstrate, whether theoretically or through a pilot study, how the adoption of KPIs can result in practical improvements in care delivery, patient outcomes, or organizational efficiency. Additionally, it's crucial to evaluate the feasibility of implementing this approach in real-world settings. Can the proposed KPIs be put into action and measured within existing healthcare infrastructures?

Thank you for this comment. As this is the first proposal of a PC-specific BSC, we do not have results of a pilot study. This has also been mentioned in the limitations section (Page 33). We have added information if the specific indicator or group of KPIs already proved to improve patient outcomes or if it is implementable within the entire results section. These informations are given together with the information on the rationale, mentioned in the previous reviewers` point.

  1. Has the proposed perspective presented in the paper been validated or tested? It is crucial to establish the credibility and applicability of the new perspective through evidence such as expert reviews, stakeholder feedback, or preliminary implementation results. The absence of such validation could significantly diminish the contribution of this paper to the field.

As this is only a first proposal of KPIs that are grouped to a novel perspective, the complete perspective itself has not been validated yet. However, all parameters within this perspective have been validated and are applicable. We inserted references on this in section 3.5 (Pages 26/27 of the revised manuscript). In addition we referred to the fact that the proposed perspective has not been validated as an entire set of KPIs yet (please see Discussion, limitations section, Page 34): ‘In particular, this is of interest for the PC-Specific Disease and Outcome perspective, that has not been prospectively validated as an entire set of KPIs yet. However, all KPIs integrated into this perspective have been externally validated on their own or in combination several times and are applicable and requestable in routine.’

5) Discussion and conclusions a. Could the authors explain how they envision applying the BSC model in healthcare practices outside of Germany, considering various healthcare landscapes, regulatory frameworks, and cultural settings? It would be beneficial to understand the factors that were considered to ensure that the model is both relevant and adaptable in different contexts.

Thank you for this comment. We have inserted a paragraph on adoption in healthcare systems outside Germany per perspective within the discussion (please see Page 33): ‘One issue of the implementation of this novel BSC is its applicability in healthcare systems outside Germany. We acknowledge that not all perspectives of the BSC are applicable. In particular nearly all KPIs of the financial perspective are only useful in the german healthcare system, besides assets or investment strategies. Within the patient and referrer perspective profitabilities and market share KPIs are useful outside Germany. However, KPIs that are associated with referrers are only applicable in countries and healthcare systems with referrers. On the other hand treatment and service KPIs are adoptable in various countries. Service and communication KPIs of the Process perspective are both not specific for one healthcare system. However, innovation and treatment indicators are not useful in centralized health care systems.

The indicators and benchmarks for the Learning and Development perspective regarding education of RP surgeons, publication performance, employee motivation and staff training are not directly related to any healthcare system, but robotic surgical systems are a prerequisite. The KPIs for employee productivity are related to the benchmarks of the German Cancer Aid. The indicators of the  PC-specific Disease and Outcome perspective are mostly derived from the German Cancer Aid and the ICHOM and are therefore applicable. ‘

  1. I suggest exploring potential obstacles to BSC adoption in clinical settings and strategies to overcome them.

Thank you very much. We have inserted a paragraph to the Discussion of the manuscript (Page 34): ‘Another issue might be adoption to clinical scenarios. Whereas KPIs of the Learning and Development phase do not only measure the performance of the individual, but are also beneficial for them, continuous monitoring of metrics like in the referrer and patient or PC-specific Disease and Outcome perspective are time consuming and challenging. Three strategies to overcome these challenges might be helpful. One option would be full integration into quality management processes. Thus users are able recognize that the integration of the BSC or KPIs is not only time consuming but can set the path on systematic quality management that is time sparing on the long run. Next, one responsible user can aid to implement the BSC to the clinic or unit. Lastly, communication of potential benefits of the entire BSC or separate perspectives and KPIs for individual employees and doctors or their education, in example in the Learning and Development or the PC-specific Disease and Outcome perspective, might help to improve acceptance of continuous quality measurements.‘

Reviewer 3 Report

Comments and Suggestions for Authors

Well written manuscript with proper scientific structure. Comprehensible English, that could be easily understood. Interesting instrument that could be utilized in prostate cancer treatment center. What was interesting for me, authors had developed described instrument on robotic prostatectomy. No other methods were included in the analysis. It could be a potential drawback and could decrease potential instrument utility, as robotic prostatectomy is still not a standard in some countries. Article will contribute to the literature of the field. However, some issues should be addressed before publication:

1.     Lines 85-86 – citation is missing 

2.     Lines 100-101 – please explain why did authors include only articles up to December 2021? It is now 2024, a lot of time has passed. Please clarify, why did you not include more up to date articles. This provides a bias to a whole investigation. 

3.     Lines 387-388 – please explain what parameters did author mind? Expand this sentence, for better context. 

4.     Lines 418 – from my point of view, this is not a proper scientific writing to include references in the conclusions section. Conclusions should summarize only author’s findings.

Author Response

Responses to Reviewer #3:

Well written manuscript with proper scientific structure. Comprehensible English, that could be easily understood. Interesting instrument that could be utilized in prostate cancer treatment center. What was interesting for me, authors had developed described instrument on robotic prostatectomy. No other methods were included in the analysis. It could be a potential drawback and could decrease potential instrument utility, as robotic prostatectomy is still not a standard in some countries. Article will contribute to the literature of the field. However, some issues should be addressed before publication:

  1. Lines 85-86 – citation is missing 

Thank you very much for reviewing our manuscript. We have included a statement to the discussion paragraph (Page 29, Line 115 and Page 29, Line 118 pp.). In addition, we have added the missing citation (Page 9): ‘Two types of metrics are distinguished in BSCs: leading indicators and lagging indicators (2,3,12).’

  1. Lines 100-101 – please explain why did authors include only articles up to December 2021? It is now 2024, a lot of time has passed. Please clarify, why did you not include more up to date articles. This provides a bias to a whole investigation. 

You are correct. The systematic review and literature search was performed beginning in the end of December 2021. Since then, we are conducting this systematic review and developed the BSC. This took approximately one and a half year. In addition, writing the manuscript took some time. We apologize for potentially having a bias here. We have included a statement on this in the discussion: ‘Next, we acknowledge that we included literature in this systematic review and BSC development until December 2021. This might cause a bias since relevant literature published in 2022 and 2023 might exist, leading to different KPIs and metrics of this BSC.’ (Page 29)

  1. Lines 387-388 – please explain what parameters did author mind? Expand this sentence, for better context. 

Our apologies for confusion. We have rephrased the sentence. In addition, we give some examples in the revised version of the manuscript (Pages 28/29): ‘Secondly, there might be some parameters that may subjectively appear equally important and worth to mention, but were not integrated. In example in the PC-specific Disease and Outcome Perspective, a normal sexual function after six months was integrated. However, there is evidence that sexual function might improve up to one and a half year after RP. Thus, some users of the BSC might prefer the metric of sexual function after one or two years. A second example might be the rate of successful catheter removal on day 5-12 after RP. However, some centers remove the catheter after three days. Thus, the KPI has to be adapted. Another example might be the utilization of the EORTC QLQ-25 questionnaire. However, other care givers might prefer the EORTC QLQ-30 questionnaire or focus more on decision regret, using the Decision Regret Score or the Brief Symptom Inventory Score. However, the EORTC QLQ-25 is reliable and common, so we decided to include this score to the BSC. ‘

  1. Lines 418 – from my point of view, this is not a proper scientific writing to include references in the conclusions section. Conclusions should summarize only author’s findings.

We have corrected this accordingly: ‘The BSC contributes to achieving the value creation envisioned by Porter's Full Cycle of Care in value-based medicine by systematically collecting and providing access to economic, personnel, and medical results, actions, and indicators.’ (Page 30).